# Enhanced Gait Recovery in Chronic Post-COVID-19 Stroke: The Role of Combined Physical Rehabilitation

**DOI:** 10.3390/reports6040051

**Published:** 2023-10-20

**Authors:** Hunor Pál Fodor, Hunor Dávid, Attila Czont, Ildikó Miklóssy, Kálmán-Csongor Orbán, Gyöngyi Tar, Abony Fodor, Zita Kovács, Beáta Albert, Pál Salamon

**Affiliations:** 1Faculty of Natural Sciences, University of Pécs, 7624 Pécs, Hungary; hunor.fodor@martonferenc.ro (H.P.F.); david_huni@yahoo.com (H.D.); czontattila@uni.sapientia.ro (A.C.); miklossyildiko@uni.sapientia.ro (I.M.); kovacszita@uni.sapientia.ro (Z.K.); albertbeata@uni.sapientia.ro (B.A.); 2Department of Bioengineering, Faculty of Economics, Socio Human Sciences and Engineering, Sapientia Hungarian University of Transylvania, 530104 Miercurea Ciuc, Romania; orbancsongor@uni.sapientia.ro; 3C. D. Nenițescu Institute of Organic Chemistry, 060023 București, Romania; 4Corax-Bioner CEU SA, 530174 Miercurea Ciuc, Romania; 5Faculty of Technical and Human Sciences, Sapientia Hungarian University of Transylvania, 540485 Târgu-Mureş, Romania; csutak@nextra.ro; 6Neurosurgery Department, Borsod-Abaúj-Zemplén County Central Hospital and University Teaching Hospital, 3526 Miskolc, Hungary; abony9@yahoo.com; 7Laboratory of Molecular Diagnostics, Emergency County Hospital, 530173 Miercurea Ciuc, Romania

**Keywords:** stroke, neurorehabilitation, combined therapy, ballistic strength training, myofascial release therapy, aerobic training, COVID-19

## Abstract

Background: Rehabilitation programs applied in cases of COVID-19-related stroke should counteract not only the effects of the stroke but also the effects of long-term COVID-19. As the molecular processes underlying these cases are still not fully understood, and evidence-based clinical outcomes are scarcely documented, there is a valid need to gather information and develop rehabilitation strategies for these patients. The risks, already clarified in the case of stroke, need to be assessed taking into account the coincidence of the two diseases. Endothelial injuries and emboli that develop after the hypercoagulable state of COVID-19 may take longer to heal, and complications may occur during exercise. This case study attempts to determine what the rehabilitation of a COVID-19-related stroke patient should include. The participant was a 64-year-old male with ischemic right middle cerebral artery stroke, left-side hemiplegia, and middle cerebral artery stenosis, and the CT showed a well-defined area of hypoattenuation in the basal ganglia territory involving the right lentiform nucleus, the anterior and posterior limbs of the internal capsule, and the dorsal part of the external capsule. His NIHSS score was 14, and he registered 15 points on the Barthel index. The patient had a COVID-19 infection two weeks before the stroke event. Methods: Conventional physical therapy was combined with adaptive ballistic strength training, a high-intensity interval training regimen, and manual treatment for myofascial release throughout the chronic recovery phase. Our primary goals were gait rehabilitation, muscle strengthening, weakness management, as well as spasticity reduction, while three different rehabilitation approaches were adopted in a single rehabilitation program to improve the outcome and long-term functional recovery of the patient. Results: The patient progressed in almost every aspect of the assessment criteria. This combined approach’s main success was improved gait speed, gait quality, and improved cardiovascular fitness. Take-away message: In the case of a stroke caused by COVID-19, where the endothelium cells are compromised, HIIT may be questionable due to the poor vascular condition. Based on our results, the low-volume HIIT approach proved appropriate and effective.

## 1. Introduction

Recovering fully after a stroke poses a substantial contemporary challenge, placing a significant strain on the healthcare system while profoundly affecting the long-term quality of life for both patients and their families. Furthermore, the process of adult neuroregeneration remains intricate, with its molecular mechanisms still not comprehensively elucidated. In cases of strokes linked with COVID-19, additional factors within the disease further exacerbate this predicament. Effective stroke prevention and rehabilitation can reduce the burden of stroke-related disability. Structured, multidisciplinary, multistage plans for effective stroke rehabilitation have been shown to improve patients’ motor recovery, general condition, and quality of life. Thus, dedicated care in multidisciplinary stroke units leads to higher independence rates from activities of daily living (ADLs) and decreases the need for long-term care after stroke [1].

Despite the fact that COVID-19 is a respiratory disease, its symptoms affect not only the lung tissue, but can have a whole range of different manifestations, such as kidney injuries, heart damage, and neurological symptoms. Neurological symptoms appear in 36% of patients and the most serious of these is stroke [2,3]. Evidence shows that a COVID-19-specific hypercoagulative state is responsible for endothelial injury, cardiogenic embolism, and, consequently, stroke, in these patients [4,5]. Moreover, the underlying complex mechanisms are still unclear, which might compromise treatment and rehabilitation strategies in these complex cases [6].

Exercise stroke rehabilitation programs are still essential for stroke survivors to reduce risk factors and morbidity linked to COVID-19’s possible long-term effects. However, there are presently no exercise rehabilitation guidelines in place for stroke survivors with a history of COVID-19 infection or with long-COVID symptoms [7].

Regardless of the setting, stroke rehabilitation must aim for the best possible outcome, both neurologically and cognitively. In this context, one of the most important factors is daily physical activity, which should be a set of exercises compiled by physiotherapy specialists and revised occasionally [8]. Relearning the lost motor function through adaptation and potentiation of neural connectivity is linked to neuroplasticity, so rehabilitation specialists aim to facilitate neuroplasticity, e.g., via constraint-induced movement therapy and task-oriented motor practice [9].

Effective neurorehabilitation strategies are based on activating experience-dependent neuronal plasticity, resulting in enhanced functional recovery after stroke. Although some principles, such as constraint-induced movement therapy, Bobath, enriched rehabilitation, VR-based rehabilitation, and exogenous or robotic interventions, already form the basis of successful rehabilitation programs, several other principles in successful neurorehabilitation have been recently identified, like multisensory stimulation, rhythmic cueing, explicit feedback/knowledge of results, motor imagery, and social interaction [10].

Evidence suggests that an integrated approach should be applied during stroke neurorehabilitation instead of triggering a limited set of neurocircuitry [11,12]. Some approaches seem to affect only specific motor areas of the cortex (e.g., massed practice, dosage, variable practice, task-specific practice, multisensory stimulation). Other approaches trigger wider networks of brain regions (goal-oriented practice, increasing difficulty, action observation, motor imagery, mirror therapy, rhythmic cueing, implicit feedback/knowledge of results, social interaction). Moreover, the effectiveness of these principles could depend on the severity of the case, the recovery phase they are applied in, and individual differences among patients [10].

Despite considerable data and evidence on successful approaches to stroke recovery, most patients do not recover sufficient motor function for an independent life despite the advances in the field. Based on data compiled by a WHO report [13], the use of rehabilitation programs can be modest, even among high-income nations, with post-stroke and cardiac rehabilitation being under-utilized even in countries like the US, Canada, or South Korea. Women and people over 65 are less likely to finish rehabilitation programs. According to a survey in Central European countries, rehabilitative services are underutilized in primary, secondary, tertiary, and community healthcare settings [13,14]. The WHO report concludes various reasons for this phenomenon, including lack of accessibility and transportation challenges, particularly for individuals living in rural regions; high service fees; protracted waiting times; and lack of awareness. Moreover, scarce infrastructural resources (equipment, space, and beds), funding gaps, and inadequacies of national legislation, regulations, or information systems are the key obstacles at the health system level [13].

Based on the above rationale, there is a stringent need for post-stroke motor function rehabilitation protocols and guidelines that can be applied in such cases, namely for patients residing in smaller towns or rural areas where specialized post-stroke rehabilitation centers are not easily accessible and state-of-the-art therapies, such as robotics or VR-aided approaches, are not readily available.

In our case study, we present a successful post-stroke recovery protocol based on a combination of methods, effective in this COVID-19-related stroke case, which can be applied in a home setting with minimal instrumentation by a skilled physical therapist. The combination of therapeutic methods consisted of conventional physiotherapy complemented during the chronic recovery phase with adaptive ballistic strength training, a high-intensity interval exercise program, and manual therapy for myofascial release.

### 1.1. Post-Stroke Rehabilitation Strategies

Dyscontrol symptoms linked to upper motor neuron (UMN) syndrome have been classified as either negative or positive, depending on whether they hinder muscle activation, cause weakness or loss of motor control, or cause stretch-sensitive muscle overactivity or spasticity [15]. As a consequence of brain injury, fewer impulses reach the motor neuron population to activate muscles or to enhance the firing rate or frequency of active motor units. This leads to decreased muscle activation and weakness, slow motor function, a loss of motor control, poor coordination, and decreased muscular activation. Although weakness directly impacts motor function, the slower build-up to peak force is also an important factor in movement slowness; thus, rehabilitation strategies should primarily focus on the negative aspects of UMN [16,17]. Nevertheless, adaptive features of UMN also need to be taken into account, as they include muscle transformation and changes in the periarticular connective tissue, which cause the muscles and connective tissue to shorten, limiting the range of motion and reducing active and passive joint mobility [18]. In prior research, gait speed has been a sensitive, valid, and reliable stroke recovery indicator representing increased mobility and functional community walking abilities [19]. It has been shown that increases in walking speed are associated with higher levels of involvement and quality of life and may reflect real mobility gains [20].

Stroke patients should generally be viewed as active learners rather than as passive collaborators in their therapy. Providing them and their caregivers or family members a participatory approach (even in the therapeutic approach) along with suitable opportunities for intense therapy and exercise, particularly to promote motor learning, plasticity, and significant improvements in general state, could be difficult in stroke rehabilitation. Recent studies emphasize the role of the personal factor and patient motivation in stroke recovery, rather than the recovery interventions limited to the narrow biomedical perspective. For example, collaborators of the Take Charge intervention trials hypothesize that a successful rehabilitation of stroke and other disabling conditions requires an acknowledgment and engagement of the “whole person” and optimizing personal motivation [21].

As post-stroke recovery-related mechanisms are time-dependent, typically several phases are distinguishable. According to the Stroke Roundtable Consortium, the acute phase would last for the first 24 h, the hyperacute phase for the first 7 days, the early subacute phase for the first 3 months, the late subacute phase for the next 6 months, and the chronic phase starting from the 6th month [22]. A broadly accepted view in stroke management states that functional recovery is strongly related to neurologic severity, and that neurologic recovery ultimately reaches a plateau phase estimated to the sub-acute and chronic phases (months 3 to 6) in the case of most patients [23].

Based on the aforementioned considerations, we anticipated that a well-adjusted recovery plan might assure functional, or even greater, neurological recovery in chronic post-stroke patients, and we give supporting results from a single-subject study to back up our hypothesis. Consequently, our approach followed international recommendations, in an outpatient setting, with individual goals and personal motivation being emphasized in intervention planning. Our intervention plan tackled each feature of the UMN, while our primary focus was gait recovery and community reintegration. The therapeutic strategy was compiled as an integrated, combined set of methods.

Our primary goals were gait rehabilitation, muscle strengthening, weakness management, as well as spasticity reduction, while three different rehabilitation approaches were adopted in a single rehabilitation program to improve the outcome and long-term functional recovery of the patient. The combined therapy was based on manual myofascial release therapy, combined with ballistic strength training and aerobic exercise. Manual therapy was applied to improve muscle tone and joint range of motion, and ballistic strength training was carried out to improve power generation and muscle strength. In contrast, the interval aerobic training enhanced cardiovascular endurance and the circulatory system.

### 1.2. Use of Manual Myofascial Release Therapy in Stroke Rehabilitation

Myofascial release (MFR) therapy is a hands-on soft-tissue technique that uses strokes with the fingers, a loose fist, the forearm, or the elbow performed at low velocity, having as its goals the relaxation of contracted muscle, increasing circulation, and venous and lymphatic drainage [24,25]. MFR can also interrupt the pain–muscle tension–pain cycle [26]. The movements have depth, duration, and direction carefully chosen to achieve the desired soft-tissue relaxation without injuring the tissue [25]. The patient can be actively involved in the stretching, generating slow movements, making the technique more effective [25]. The relaxation of soft tissue may be the result of neuro-reflexive changes. Receptors in the stretched tissue receive afferent stimulation, which, after central (spinal cord, cortical) processing, can cause efferent inhibition [27].

Tui Na (Chinese massage) seems to be effective in improving upper and lower limb motor function as well as in subacute post-stroke spasticity management [28], especially of the elbow flexors, wrist extensors, and knee flexors and extensors [29].

Myofascial structural integration positively impacted spastic cerebral palsy children’s Gross Motor Function Measure scores [30].

MFR performed with a tennis ball improved the balance of chronic spasticity patients [31], but also gave promising results for improving the spasticity of muscles and upper limb motor functions in stroke patients [32].

A comparative study [27] investigated the effects of the tendinous pressure technique and manual myofascial release and they found that both are effective in treating spasticity following stroke; however, tendinous pressure gave better results in muscle reaction testing. MFR improved the motor function of children with moderate to severe spastic cerebral palsy [26]. MFR performed alone or alongside conventional treatment reduced lower limb spasticity in diplegic cerebral palsy [33,34], and reduced spasticity [35] and increased the ROM of lower extremity joints [36] in chronic spastic conditions after stroke.

### 1.3. Adaptive Ballistic Strength Training Exercises in Neurorehabilitation

Over the last two decades, it has become increasingly accepted that the key problem limiting mobility in upper motor neuron syndrome is muscle weakness or paresis [37]. The optimal strengthening protocol is not determined [38,39], but there are strong recommendations that strength training is effective, although there is little overlap/evidence for gait development/improvement [40,41]. Williams et al. investigated task specificity of strength training for walking in neurological conditions and found quadriceps and hamstring exercises to be the most commonly used exercises in neurological rehabilitation. However, these are not task-specific in the clinical population [40].

### 1.4. Aerobic Training in Stroke Recovery

A growing body of data on this topic provides clear evidence that aerobic exercise is more effective than conventional treatments in post-stroke motor function recovery. Rehabilitation protocols that include aerobic exercise or mixed exercise regimes with an aerobic component improve both functional mobility and cardiovascular fitness in patients with chronic stroke. Therefore, it can lead to better outcomes than conventional rehabilitation protocols [42,43,44].

Rehabilitation protocols for stroke survivors should prioritize muscle-strengthening exercises, low- to moderate-intensity aerobic exercise, increasing physical activity/minimizing inactivity, and risk management for secondary stroke prevention. Aerobic training improves chronic stroke patients’ quality of life and the mortality index [44]. It has been shown that aerobic training improves cardiorespiratory fitness, muscular endurance, and functional recovery in stroke patients [45]. Studies and guidelines from as early as 2013 [43,46,47] recommend incorporating aerobic exercise into routine post-stroke care.

## 2. Detailed Case Description 

### 2.1. Patient Characteristics: De-Identified Patient-Specific Information

The patient under consideration is a 64-year-old male with an extensive history of good health and no chronic medical conditions. Notably, he had not received vaccination against SARS-CoV-2.

Prior to the onset of the stroke, the patient maintained an active and dynamic lifestyle. His dietary habits were characterized by a diet predominantly consisting of wild game he personally hunted, such as deer and boar, as well as locally sourced fish. His diet was notably rich in lean proteins and omega-3 fatty acids, which are often associated with potential cardiovascular benefits. This dietary pattern reflected a lifelong affinity for outdoor activities and hunting, underscoring his deep connection to nature and its resources.

In addition to his dietary choices, it is important to note the patient’s alcohol consumption. He regularly consumed spirits in moderate quantities, typically approximately 2 centiliters of spirits daily, prior to the main meal of the day. He also indulged in social wine consumption, partaking in such occasions approximately twice a week. During these social gatherings with friends, he consumed an estimated 0.8 L of wine on each occasion.

It is crucial to acknowledge that these dietary and alcohol consumption patterns were integral to his daily life and social interactions. They represented aspects of his lifestyle that pleased him, formed part of his cultural identity, and contributed to his overall well-being.

The combination of an active lifestyle, a diet rich in lean proteins and omega-3 fatty acids, and moderate alcohol consumption characterized the patient’s life before the stroke event. When considered in the context of his medical history, these factors provide valuable insights into his pre-stroke health and lifestyle choices.

### 2.2. Patient Characteristics: Primary Concerns and Symptoms of the Patient

Two weeks before the stroke event, the patient contracted a COVID-19 infection. He experienced symptoms consistent with the virus, including shortness of breath, persistent fatigue, fever, and a cough. These symptoms, indicative of a respiratory infection, were managed at home. However, despite the severity of his symptoms, he expressed a strong reluctance to seek medical attention at a healthcare facility.

This resistance to hospitalization during his COVID-19 infection emphasized his aversion to healthcare settings and his preference for managing health challenges independently. It is important to address this aspect of his medical history as it may have contributed to his post-stroke care decisions and overall approach to health management.

On the day of the stroke onset, the patient experienced significant speech difficulties, rendering him unable to complete a sentence, and he also noted unilateral weakness. Subsequently, an ambulance was summoned, and, upon their arrival, the medical team presented the patient with the option to proceed to the hospital for further evaluation or remain at home. The patient ultimately decided to stay at home. However, later in the evening, he suddenly realized that his left side was completely paralyzed when attempting to get up to use the restroom.

### 2.3. Medical, Family, and Psychosocial History, including Relevant Genetic Information

On the paternal side of the patient’s family, there was a noteworthy prevalence of stomach-related issues, suggesting a potential genetic predisposition or familial tendency toward gastrointestinal conditions. Although the specific nature of these stomach issues was not elucidated, they formed an integral part of the patient’s familial medical background.

Conversely, the maternal side of the family displayed a distinctive pattern of lower limb circulation problems and brain atrophy. These familial traits indicated potential genetic factors contributing to vascular health and neurodegenerative conditions. Lower limb circulation problems could suggest a susceptibility to vascular diseases, while the presence of brain atrophy within the family hinted at a possible predisposition to neurological disorders.

The pre-hospital assessment showed a left facial drop, slurred speech, coordination deficits, and left motor hypotonia.

On the day of the participant’s urgent transport to the hospital, he was diagnosed with an acute ischemic stroke involving the right middle cerebral artery territory, specifically identified as i69.3 in the International Classification of Diseases (ICD-10). This diagnosis was based on clinical evaluation and the findings from a non-contrast head CT scan (Figure 1). Non-contrast head CT showed a well-defined area of hypoattenuation in the basal ganglia territory involving the right lentiform nucleus, the anterior and posterior limbs of the internal capsule, and the dorsal part of the external capsule. The rest of the cerebrum was preserved. There was no sign of acute bleed, and no significant mass effect was seen. (Overall features were those of acute partial right lateral lenticulostriate infarction).

After the stroke, the patient was in hospital care for 14 days (9 October 2021–23 October 2021), where the main points of movement rehabilitation were early mobilization with passive movements, like positioning in bed in different postures (lying on the side, lying on the stomach); increasing alertness and improving sitting balance by sitting up in bed and sitting out of bed; and sensory stimulations of the auditory, olfactory, visual, tactile–kinesthetic and vestibular systems. At discharge, the patient was incapable of walking or changing his body position, his NIHSS (National Institutes of Health Stroke Scale—measuring the severity of stroke) score was 14, and he registered 15 points on the Barthel index. He displayed no sensory deficits, and his visual fields were intact.

### 2.4. Experimental Design

The patient was enrolled in the experiment after his discharge from the hospital on 23 October 2021. The baseline was assessed before rehabilitation for two weeks, with weekly assessment. The movement rehabilitation process was carried out at the patient’s home, and the goals were set with the patient and his wife. Standard assessment methods were applied at the end of rehabilitation phases, and data were collected and visualized.

### 2.5. Physiotherapy Assessment

Standardized assessment methods were applied to determine patient status at the end of each phase. In contrast, a baseline assessment was performed during the first two weeks after hospital discharge. Functional and stroke-specific assessment was carried out based on the National Institutes of Health Stroke Scale (NIHSS), Barthel index, Manual Muscle Testing, Modified Ashworth Scale, 10-meter walk test, 6-minute walk test, and active range of motion (AROM) with goniometry, following standard protocols.

The National Institutes of Health Stroke Scale (NIHSS) exhibits robust reliability, particularly regarding inter-rater consistency, signifying that different healthcare professionals should yield similar scores when evaluating the same patient. Nevertheless, achieving this high level of agreement among raters may necessitate some training. Additionally, the NIHSS demonstrates commendable validity in both content and construct aspects, effectively capturing various dimensions of stroke severity, such as motor function, language, and consciousness. It has earned widespread adoption in both clinical practice and research settings. Furthermore, the NIHSS displays sensitivity to changes in stroke severity over time, making it a valuable tool for monitoring a patient’s progress during rehabilitation or post treatment, enhancing its utility in the healthcare continuum [48].

The Barthel index is a reliable tool for assessing a patient’s ability to perform activities of daily living (ADLs), often demonstrating good inter-rater reliability due to its straightforward nature and well-defined scoring criteria. In terms of validity, it has good face validity as it measures crucial ADLs like bathing, dressing, and mobility. However, it may not encompass all facets of functional ability, and its sensitivity to mild functional impairments can be limited due to a ceiling effect. Nevertheless, the Barthel index is sensitive to changes in functional status. It is a valuable instrument for assessing rehabilitation outcomes, particularly in stroke patients, where monitoring improvements in ADLs is crucial [49].

The Modified Ashworth Scale, employed for evaluating muscle tone and spasticity, exhibits commendable inter-rater reliability, particularly when administered by trained assessors. Nonetheless, it is noteworthy that this reliability can be influenced by the level of experience of the examiner and the patient’s positioning during the assessment. In terms of validity, the scale possesses sound face validity for the appraisal of muscle tone and spasticity, which are prevalent concerns in the context of stroke patients. Nevertheless, it should be acknowledged that the scale may not comprehensively encompass the multifaceted nature of muscle tone abnormalities. Importantly, the Modified Ashworth Scale demonstrates sensitivity to alterations in muscle tone and spasticity across time, rendering it a valuable instrument for the monitoring of treatment efficacy within stroke rehabilitation and related clinical contexts [50].

Manual Muscle Testing, a method employed to assess muscle strength, exhibits commendable inter-rater reliability, particularly when administered by proficient evaluators. The utilization of standardized positions and testing protocols bolsters this reliability. In terms of validity, the method demonstrates sound content validity as it directly quantifies muscle strength. Nonetheless, it is pertinent to acknowledge that Manual Muscle Testing may not fully encapsulate the intricacies of functional limitations arising from muscle weakness during complex movements. Crucially, this assessment technique displays a heightened sensitivity to fluctuations in muscle strength over time, rendering it a valuable tool for the evaluation of muscle recovery in stroke patients and other clinical scenarios where muscle strength assessment is pertinent [51].

The assessment of active range of motion (AROM) through goniometry represents a reliable method. However, variability in measurements may arise due to factors such as patient discomfort or their level of co-operation during the evaluation process. In terms of validity, AROM goniometry demonstrates validity as it directly quantifies the extent of joint mobility, thereby discerning limitations in joint range of motion that could influence functional activities. Importantly, this assessment modality exhibits a heightened sensitivity to fluctuations in joint range of motion over time, rendering it an invaluable tool for systematically monitoring improvements in this domain during rehabilitation interventions [52].

The 10-m walk test (10MWT) assessment is frequently employed to evaluate gait and walking speed among individuals recovering from strokes. It serves as a valuable tool in clinical practice and research settings, enabling the systematic examination of stroke patients’ walking abilities and pace, thus contributing to a comprehensive understanding of their mobility and recovery progress [53].

The 6-min walk test (6MWT) is a widely adopted clinical assessment that quantifies the distance an individual can ambulate within a span of 6 min. It is frequently employed to evaluate functional endurance and cardiovascular fitness in various patient populations, providing valuable insights into a person’s physical capacity for sustained activity and aerobic performance. This test’s utility extends to clinical practice and research, offering a standardized and objective measure of functional capacity to aid in treatment planning and outcome assessments [54].

### 2.6. Therapeutic Interventions

In phases I-III, conventional physiotherapy was completed using a combined therapeutic approach in phase IV.

The patient received a 24-week conventional movement therapy described in Table 1.

The rehabilitation program starting from phase IV for the patient included 3 main components: (1) myofascial release, (2) ballistic strength exercises, and (3) high-intensity interval training (Table 2). The patient achieved independence in dressing and self-care routines, adapted to his needs. He developed the ability to navigate stairs step by step, ensuring safety. However, his endurance did not significantly improve, as medium-distance walks of approximately 500 m were still challenging. Furthermore, the quality of his gait remained suboptimal, characterized by proximal compensation mechanisms and reduced power generation in the distal extremities.

### 2.7. Statistical Methods

Due to the single-subject design of the current investigation, statistical analysis was not used. Unless otherwise stated, the outcome measures are shown as raw values and changes between phases in each case. When possible, we present comparisons with existing data, such as minimally detectable change.

### 2.8. Results

In our case study, we describe a successful post-stroke rehabilitation program based on a variety of techniques that may be used at home by qualified physical therapists with little instrumentation. Conventional physical therapy was combined with adaptive ballistic strength training, a high-intensity interval training regimen, and manual treatment for myofascial release throughout the chronic recovery phase, as follows:

Phase I—hypotonic phase, 2–8 weeks: Conventional movement therapy was carried out where the main rehabilitation goal was preventing abnormal movement patterns, establishing a controlled movement of the upper and lower trunk, and improving bed mobility. We applied 40 min sessions four times a week for weeks 2–4, and three times weekly 50 min sessions for weeks 4–8. After phase I of conventional physiotherapy, NIHS scale improved for left leg motor drift and facial palsy (Table 3), but due to muscle weakness limb ataxia was not testable, and the domain of activity of daily living shows improvements in bowel control and transfers (bed to chair and back). The assessment of lower limb muscle strength became possible: the patient performed some effort against gravity, spasticity developed, and the Modified Ashworth Scale showed a score of 2 for proximal muscle groups and 3 for ankle plantar flexors. The assessment of active range of motion with a goniometer was developed to a greater extent in the hip joint and to a smaller extent in the knee and ankle joints (Table 4). Assessing gait velocity and walking endurance was not feasible.

Phase II—hypertonic phase, 9–16 weeks: The patient developed muscle spasticity. Thus, our goal was spasticity reduction, static and dynamic balance improvement, independent toileting, and mobility in the apartment. Conventional therapy was applied three times weekly in 50 min sessions. After phase II of conventional physiotherapy, the NIHS scale improved for left leg motor drift (Table 3), and a slight improvement in left arm motor drift was registered (Table 4). Due to muscle weakness, limb ataxia was not testable, while retesting the Barthel index (which increased from 25 to 60) showed higher scores in toilet use, bladder control, transfers, and gait. A significant improvement in performing short-distance walks due to lower limb muscle strength was achieved. Muscle spasticity was reduced and active range of motion was increased in this phase. In addition, the patient managed to perform the 10MWT to assess walking speed. In phase III, the patient managed to perform the 6MWT (Table 5), but no significant improvements in muscle strength, spasticity, or gait speed could be registered. However, a slight improvement in the domain of activity of daily living was registered, resulting in the patient being able to perform grooming.

Phase III—late subacute phase, 17–24 weeks: The patient reached a plateau phase, muscle spasticity did not improve, weak endurance, gait quality stagnated with proximal compensation, and distal power generation reduced. Phase goals were set to spasticity reduction, range of motion improvement, achieving medium distance safe walks of 500 m and improved gait quality, stair use, and grooming and dressing. Fifty-minute sessions of conventional physiotherapy were applied three times weekly. After the third phase, we reached a stage of our rehabilitation where the patient’s development stagnated and did not improve, so it was necessary to change the therapeutic interventions to achieve the goals set out in the third stage.

Phase IV—multiple therapeutic approach phase, 25–32 weeks after discharge: Improved active range of motion and strength, positive change in VO_2_ max, and improved gait velocity were observed. Phase goals were set to achieve medium-distance safe walks of 500 m, improved gait quality, and better cardiovascular fitness. In this phase, a combination therapy was applied, consisting of myofascial release—MFR—carried out once a week in 50 min sessions, adapted ballistic exercise four times/week in 15–20 min sessions, and high-intensity interval training—HIIT—20 min sessions applied four times a week. In the chronic rehabilitation phase (weeks 25–32), however, the patient progressed in almost every aspect of the assessment criteria using a combined set of rehabilitation methods. The patient’s goal was to achieve intermediate 500 m walks without fatigue at a faster pace (0.30–0.35 m/s). To achieve this, we changed the rehabilitation plan: manual therapy (myofascial release) was used to normalize spasticity and range of motion, adapted ballistic strength exercises for power generation, and high-intensity interval training to improve cardiovascular endurance. As a result, a 0.43 m/s walking speed was achieved at the end of phase IV (10MWT, see Table 5) and registered a 127 m value on the 6MWT (Table 5).

A schematic outline of the recovery strategy and the results achieved is presented in Figure 2.

Two weeks after discharge, the NIHS scale showed improvement at the levels of consciousness, sensation, language, dysarthria, and extinction/inattention, but due to muscle weakness limb ataxia was not testable. The Barthel index score was unchanged (Table 3), and assessments of muscle strength, spasticity, active range of motion (Table 4), gait velocity, and walking endurance were not feasible due to the patient’s general status.

Throughout the rehabilitation process, the patient underwent a single medical examination during which brain imaging scans were not conducted due to the context of home-based rehabilitation. While this limitation affected the availability of certain diagnostic data, an analysis of functional recovery between the admission and chronic stages provides valuable insights.

Notably, after undergoing conventional physiotherapy, the patient demonstrated significant improvements across various assessment parameters, including Manual Muscle Testing, Modified Ashworth Scale, active range of motion (AROM) measured using goniometry, and the 10-meter walk test (10MWT). At this juncture, the patient was able to perform a 6 min walk test, albeit with intermittent pauses and limited coverage (approximately 85 m). This transitional phase suggested an initial plateau in functional improvement.

Subsequently, following the implementation of our combined physical rehabilitation approach, noteworthy advancements were observed. These encompassed improvements in Manual Muscle Testing, Modified Ashworth Scale scores, enhanced active range of motion (AROM) as measured by goniometry, superior performance in the 10-m walk test (10MWT), and a substantial improvement in the 6 min walk test distance. This progress in functional metrics was also indicative of enhanced aerobic capacity.

Furthermore, the patient reported a meaningful increase in social engagement, evidenced by participation in various cultural and recreational events. This multifaceted improvement not only underlines the efficacy of the combined rehabilitation approach but also highlights the tangible impact on the patient’s overall quality of life and social participation.

## 3. Discussion

The gold standard in clinical research is considered to be the randomized parallel-group clinical trial design, which provides statistics-validated results for the outcome of a given treatment or drug on a specific, carefully selected population. Randomized controlled trial (RCT) designs usually work with large sample numbers and involve strictly controlled protocols. At the same time, they could miss variations over time due to a limited number of measurements or simply miss the individual responses of patients, both of which contribute to the generally accepted principle of personalized medicine.

There are situations when these results might not be applicable in determining the most effective treatment for an individual patient because of individual clinical features or the heterogeneity of populations usually participating in clinical trials, or because the strict selection criteria for subjects entering clinical trials might limit the general applicability of the observed results [55].

HIIT has proven to be very effective in post-stroke clinical populations [56,57,58]. The protocols can be classified into three main types depending on high-intensity burst duration, recovery duration, and recovery type [59]. Short-interval HIIT consists of supramaximal high-intensity bursts, 15–60 s, and a 1:1 burst-to-recovery ratio. Low-volume HIIT applies short, high-intensity bursts of 10–60 s, at a maximal absolute or anaerobic workload, and burst-to-recovery ratios of 1:2 or 1:4 min, while in the case of long-interval HIIT, high-intensity bursts (3–4 min) at lower workloads (80–90%) and a burst-to-recovery ratio of 1:1 or 4:3 with active recovery are applied. The cardiovascular training parameters of stroke survivors are still not defined, although it is known that the exercises induce profound changes in brain structure. However, in the case of a stroke caused by COVID-19, where the endothelium cells are compromised, HIIT may be questionable due to the poor vascular condition. A sudden increase in blood pressure during HIIT can cause another cerebrovascular event. While this may be of concern, whether the degree of hypertensive response induced with high-intensity exercise is potentially damaging has not been confirmed. Thus, one of our goals was to set up a safe and effective protocol for HIIT training in the case of our stroke patient. Overall, based on our results, the low-volume HIIT approach and the protocol we set up proved to be appropriate and effective.

Evidence-based practice (EBP) can profit from this type of trial by providing clinical practitioners with practical information to support decision-making within a specific clinical environment; in our case, this was by providing home-based, easily accessible care for an elderly patient. However, as the greatest limitation of this type of study design, changes in the monitored variables cannot be attributed clearly to the intervention due to a lack of statistical backup.

In phase IV, we chose three therapeutic interventions, taking into account the specifics of COVID-19-related stroke risks, to improve cardiovascular fitness, gait velocity, and gait quality. First, we wanted to prepare the muscle tissue and joints for better movement quality, so we applied for myofascial release once a week.

MRF therapy is thought to normalize the length and sliding properties of restricted myofascial tissue, restoring joint mobility to some extent [32,60,61]. We selected MRF therapy for our combined approach using these considerations and hypothesized that it had a facilitating effect on both ballistic training exercises and HIIT program completion by the patient. Although the discrete effects of MRF therapy can be difficult to distinguish in such a complex rehabilitation case, we think that improvements observed in the AROM scores during the chronic phase of recovery (ankle plantar flexion improvement from 36° to 45° and ankle dorsiflexion from 6° to 8°, respectively) can partially be attributed to the applied MRF therapy.

At the same time, we started power training of the affected lower limb by adapting ballistic exercises to improve gait speed and achieve better power generation for gait recovery. Our patient performed adapted high-intensity interval training in this rehabilitation phase to improve cardiovascular endurance. The concept was to prepare the muscles and joints with myofascial release to perform ballistic exercises and HIIT to achieve our goals.

Strength training is applied for gait improvement for a variety of neurological conditions. According to a systematic review [25], however, based on an investigation of 25 clinical trials, only 10 resulted in significant strength improvement; among these, only 3 studies reported on significant gait improvement, while in 2 of the 3 studies patients received integrated stroke rehabilitation interventions (including balance training and aerobic training). These findings further support our approach of applying an integrated intervention plan and focusing on the specificity and biomechanics of gait training (with emphasis on the muscles responsible for power generation and high angular velocity). The patient gained propulsion power by performing ballistic exercises, resulting in reduced proximal compensation. Performing HIIT with reduced proximal compensation significantly helped the patient to achieve his goals, while the applied low-volume HIIT approach proved to be safe and effective in the framework of this single case study for COVID-19-related stroke.

Notable improvements were seen once our combined physical rehabilitation strategy was put into practice. These included increased active range of motion (AROM) as determined via goniometry, improvements in Manual Muscle Testing and Modified Ashworth Scale scores, greater performance in the 10-meter walk test (10MWT), and a significant increase in the 6 min walk test distance.

Although all three components of this program may influence endurance and gait speed, the focus of this article is the conceptual basis of multidisciplinary therapy improving gait quality (by lowering proximal compensation) and cardiovascular fitness, and the main success of this combined approach could be the improvement in the overall quality of life of the patient, significantly influencing re-integration and leading to a socially active lifestyle.

## 4. Conclusions

In this section, we aim to shed light on our patient’s personal perspective and the impact of his medical condition and treatment on his daily life and goals. The patient expressed a strong desire to regain his ability to enjoy medium-distance outings and social activities with his wife, which had been a significant source of joy and connection for them.

Prior to his medical condition, the patient and his wife regularly enjoyed going to the cinema, attending live theater performances, joining meetings for hunters, and leisurely shopping trips together. However, following his diagnosis and subsequent treatment, our patient found these activities to be physically challenging and mentally draining due to his limited mobility and fatigue.

Our patient’s determination to achieve his goal of resuming these activities remained a central driving force throughout his rehabilitation journey. With the guidance and support of his healthcare team, he worked diligently on his multimodal physical therapy exercises. Over time, he regained his strength, balance, and confidence.

One of the significant milestones for our patient was the day he and his wife were able to return to the cinema for the first time in months. He described this experience as emotionally uplifting and a testament to the progress he had made in his recovery. Gradually, he also resumed attending theater performances and accompanying his wife on shopping excursions and joined meetings for hunters.

The patient’s journey reflects the broader impact of his combined physical rehabilitation treatment, highlighting not only the clinical aspects but also the human aspects of recovery. It underscores the importance of patient-centered care, where the patient’s own goals and experiences are considered integral to the overall management of their condition.

Present evidence from our case study seems to support our hypothesis that a well-adjusted recovery plan could guarantee functional or even greater neurological recovery, even for patients who do not have access to state-of-the-art stroke rehabilitation centers. Moreover, our study may contribute to the assessment of HIIT training application in COVID-19-related stroke rehabilitation programs. In the outpatient setting, our strategy adhered to international recommendations, emphasizing personal motivation and individual goals in intervention planning. Our intervention strategy addressed every aspect of the UMN, but our main priorities were gait improvement, muscle strengthening, managing weakness, and reducing spasticity. While our results support the effectiveness of a tailored rehabilitation plan, including low-volume HIIT, for improving functional outcomes in COVID-19-related strokes, further research is needed to better understand the complex interplay of vascular conditions in such cases.

## Figures and Tables

**Figure 1 reports-06-00051-f001:**
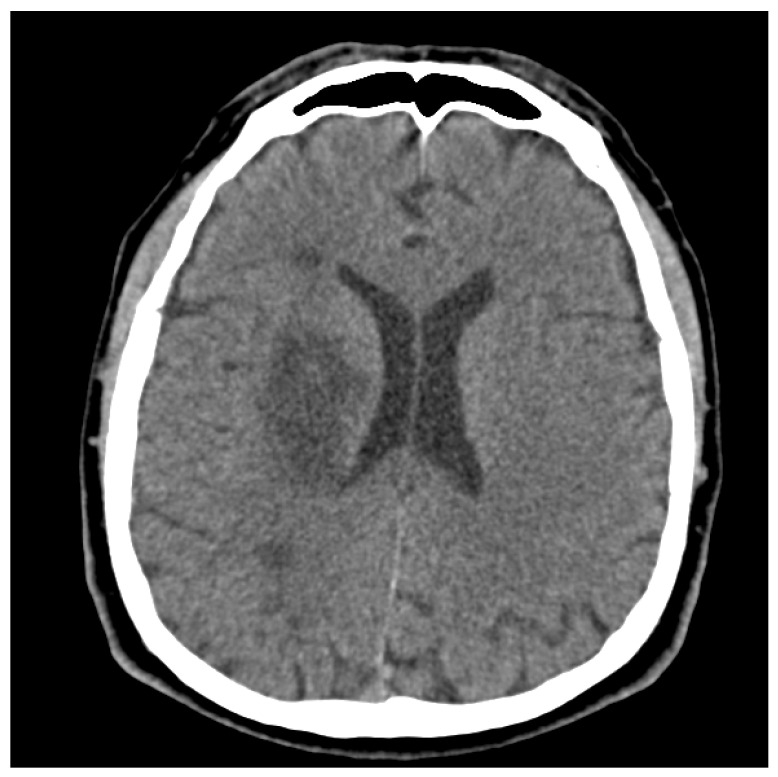
Non-contrast head CT scan of the patient on the first day of hospitalization.

**Figure 2 reports-06-00051-f002:**
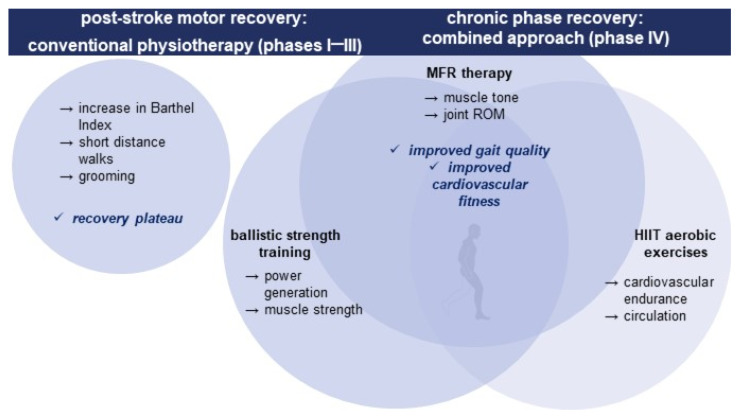
Post-stroke motor recovery strategy and summary of intervention results.

**Table 1 reports-06-00051-t001:** Conventional movement therapy interventions.

Phase	I. (Week 2–8)
Patient’s status	Left motor hypotonia; incapable of walking or changing body position; National Institutes of Health Stroke Scale (NIHSS) score 14; Barthel index score 15
Dosing	4 times/week for 40 min (week 3–5); 3 times/week for 50 min (week 6–8)
Goals	Preventing the development of movement patterns resulting from abnormal muscle tone; establishing controlled movement of the upper and lower trunk; bed mobility: scooting up, scooting down, scooting laterally, rolling in bed, supine to sitting, sitting to supine; transfers (bed to chair and back)
Exercises	Circulatory and breathing exercises; passive movement (particular attention to the hip and shoulder joints); assisted active movement; rhythmic initiation, Proprioceptive Neuromuscular Facilitation (PNF) technique; active internal and external rotation of the hip; hip adduction exercises; “bridging” (hip extension); sequence of exercise progression following the patterns of motor development acquired during the infant’s life (rolling, sitting); adapted progressive resistance and isometric strengthening exercises to strengthen the trunk muscles to stabilize sitting position; “rhythmic stabilization” and “alternating isometrics” PNF techniques to fix sitting
Outcome	The goals set for the first phase of our rehabilitation were achieved.

**Phase**	**II. (Week 9–16)**
Patient’s status	Spasticity had already developed on the affected side, pronounced in the upper limb; shoulder drawn backward, arm turned inward, elbow bent with fisted hand and palm down, pelvis drawn backward with the leg turned inward, hip, knee and ankle straightened, foot stiffened downward
Dosing	3 × 50 min per week
Goals	Reduction in spasticity, normalization of muscle tone; improvement in static and dynamic balance; independent toileting; mobility in the apartment with a 4-point cane
Exercises	“hold and relax” and “rhythmic rotation” PNF technique for increased muscle tone; stretching under load for tone reduction; range of motion (ROM) exercises; adapted progressive resistance training and isometric exercises for strength training of the trunk and lower limbs; static and dynamic balance exercises, and “rhythmic stabilization” PNF technique; assisted walking exercises with weight reduction by therapist, step variations, tandem walking, side-step walking, backward walking
Outcome	The goals of the second phase of our neurorehabilitation were achieved. Reduced spasticity, improved lower limb strength, and static and dynamic balance resulted in walking with a cane in the apartment.

**Phase**	**III. (Week 17–24)**
Patient’s status	Hypertonia on the left side, shortened lateral trunk.
Dosing	3 × 50 min per week
Goals	Reducing muscle stiffness; maintaining ROM; improving gait quality by lowering proximal compensation and improving distal power generation; grooming, dressing, stair use (16 steps); achieving medium-distance safe walks of 500 m
Exercises	Spasticity reduction, muscle tone normalization with stretching under load, PNF techniques and ROM exercises; gait training with static and dynamic balance exercises; 15 min of adapted progressive resistance training for strengthening lower limb muscle groups: leg extension, leg curl, sit to stand, step ups
Outcome	The goals of the third phase were achieved to a limited extent: our patient has learned how to dress and wash himself in an adapted way; goes up and down the stairs by step to step; his endurance has not improved, medium-distance walks (500 m) were not feasible; the quality of gait has not improved, there remains proximal compensation and reduced distal power generation

**Table 2 reports-06-00051-t002:** Combined rehabilitation interventions.

Phase	IV. (Week 25–32)		
Therapeutic intervention	**Myofascial release (MFR)**	**Ballistic strength training (BST)**	**High-Intensity Interval Training (HIIT)**
Dosing	Once a week for 8 weeks, 50 min/session	4 times a week, 15–20 min/session	4 times per week 20 min/session
Goals	Breaking down tissue adhesions; improving ROM; reducing spasticity	Improving the three main areas of power generation for propulsion; improving lower limb strength	Improving cardiovascular endurance; improving gait speed
Exercises	30 s of superficial myofascial spreading and 3–5 min of slow strokes (with fingers, loose fists, elbow or forearm adjusted to local myofascial tissue barriers and relaxing ability) on plantar fascia; calf muscles; hamstrings; superior and inferior extensor retinaculum; tibialis anterior/extensor hallucis longus	Exercise 1: ankle plantar flexion, maximum power and velocity 2 × 8 repetition 1 min interserial recovery, 3 s intraserial recovery (0–4 week), same exercise with elastic band resistance (5–8 week) Exercise 2: patient in standing position in hip flexion 90 grade, hip; extension with maximum power and velocity, 2 × 8 repetition. 1 min; interserial recovery, 3 s intraserial recovery (0–4 week), same exercise; with elastic band resistance (5–8 weeks) Exercise 3: patient in standing position, hip flexion over 90 grade, maximum power and velocity, 2 × 8 repetition, 1 min interserial recovery, 3 s intraserial recovery (0–4 week), same exercise with elastic band resistance (5–8 week); 3 min recovery between exercises	Maximum safe speed walking (0.33–0.40 m/s) for 30 s followed by 2 min of recovery, a total session time of 20 min
Outcome	Improved ROM, reduced spasticity; improved lower limb strength and power generation; improved gait speed; improved cardiovascular endurance		

**Table 3 reports-06-00051-t003:** Evolution of stroke-specific indexes during rehabilitation.

	Acute Phase	Early Subacute Phase		Late Subacute Phase		Chronic Phase
	Hospital assessment	Baseline	End of Phase I	End of Phase II	End of Phase III	End of Phase IV
		Weeks 0–2	Weeks 3–8	Weeks 9–16	Weeks 17–24	Weeks 25–32
National Institutes of Health Stroke Scale (NIHSS)	14	9	6	4	4	4
	untestable limb ataxia					
Barthel index	15	15	25	60	70	80

**Table 4 reports-06-00051-t004:** Evolution of muscle strength, muscle tone, and active range of motion of the affected side during the investigation period.

		Early Subacute Phase		Late Subacute Phase		Chronic Phase
		Baseline	End of Phase I	End of Phase II	End of Phase III	End of Phase IV
		Weeks 0–2	Weeks 3–8	Weeks 9–16	Weeks 17–24	Weeks 25–32
Manual Muscle Testing on affected side (MMT)	Hip flexors	0	3−	3+	3+	4+
	Hip extensors	0	3	4−	4−	4
	Knee flexors	0	2	2+	2+	3+
	Knee extensors	0	2	3	3	4−
	Ankle plantar flexors	0	3−	3+	3+	4
	Ankle dorsal flexors	0	0	2−	2+	2+
Modified Ashworth Scale on affected side	Hip internal rotator	-	2	1+	1+	1
	Knee extensors	-	2	1+	1+	1
	Ankle plantar flexors	-	3	2	2	1+
Active range of motion (AROM) via goniometry	Hip flexion	0°	88°	96°	98°	108°
	Hip extension	0°	10°	14°	15°	20°
	Knee flexion	0°	15°	88°	90°	120°
	Ankle plantar flexion	0°	22°	35°	36°	45°
	Ankle dorsal flexion	0°	0°	2°	6°	8°

**Table 5 reports-06-00051-t005:** Gait velocity, endurance, and aerobic capacity testing.

	Early Subacute Phase		Late Subacute Phase		Chronic Phase
	Baseline	End of Phase I	End of Phase II	End of Phase III	End of Phase IV
	Weeks 0–2	Weeks 3–8	Weeks 9–16	Weeks 17–24	Weeks 25–32
10-meter walk test (10MWT) fast walking speed with 4-point cane	0 m/s	0 m/s	0.28 m/s	0.31 m/s	0.43 m/s
6-minute walk test with 4-point cane (with supervision)	-	-	-	85 m, VO_2_ peak: 15.8	127 m, VO_2_ peak: 16.47

## Data Availability

Data are contained within the article.

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
