# Peer review of "Enhanced Gait Recovery in Chronic Post-COVID-19 Stroke: The Role of Combined Physical Rehabilitation"

_reports, 2023, doi:10.3390/reports6040051_

Round 1

Reviewer 1 Report

A combined physical rehabilitation to enhance gait recovery in a chronic post-stroke patient was described in this manuscript. The approach is interesting, but there are some important issues that must be addressed:

- Since this is a clinical case, I recommend authors follow the CARE guidelines to write the manuscript.

- Nothing is explained in relation to ethical aspects. Does this study have the approval of an Ethics Committee? Was the patient informed of the study and did he/she sign consent to participate?

- The methods section does not explain the evaluation measures, nor their psychometric characteristics in stroke.

Minor issues:

- I recommend reorganizing the introduction to make it easier to follow the information presented. For example, the first paragraph about COVID seems out of place to me. I would first introduce the information on "Post-stroke rehabilitation strategies" and then specify the three included in the combined therapy of the case presented.Furthermore, it would indicate whether there is evidence of the use of Myofascial release (MFR) therapy in people with stroke or in neurological patients.

- I recommend improving Table 1 to make it more visual. Similarly, Table 2 should also be reviewed.

- The first 5 paragraphs of the results section practically repeat the intervention without providing results.

- Regarding the conclusions, I recommend rewriting them since, except for the last paragraph, the rest repeats information or is a poorly founded conclusion as it is based on a single clinical case. In the last paragraph they talk about "community reintegration" when this result has not been studied.

I recommend reviewing the English and simplifying the sentences to facilitate understanding. For instance, see lines 87-89 and 89-91.

Author Response

Dear Reviewer,

We extend our gratitude for your meticulous review of our manuscript and for providing valuable insights and recommendations for its enhancement. Your expertise and careful evaluation are sincerely appreciated.

We are pleased to respond to your comments and suggestions, addressing each point you raised:

Q: Since this is a clinical case, I recommend authors follow the CARE guidelines to write the manuscript.

A: Thank you for your suggestion, the manuscript text was updated according to CARE guidelines considering the relevant aspects of our case report. E.g. the abstract was updated accordingly with structured information about case specifics, the most important clinical findings regarding the patient, and a summary of the applied interventions and conclusions, see lines 17-44. We appreciate your thorough review, regarding the patient information section. We would like to inform you that we have completed the patient information section, addressing the patient's primary concerns and symptoms, along with their medical, family, and psyhosocial history, including any pertinent genetic information, see lines 230-298. However, it is important to note that there were no relevant past interventions to report for this particular patient. 

Q: Nothing is explained in relation to ethical aspects. Does this study have the approval of an Ethics Committee? Was the patient informed of the study and did he/she sign consent to participate?

A: Thank you for your inquiry and question regarding the ethical aspects. We would like to assure you that the ethical considerations of our study were rigorously adhered to. The necessary ethical approval for our study was obtained from the relevant Ethics Committee. The exact designation of the approval and details of the approving authority can be found in the appropriate section of our paper, lines 622-625. In addition, the participant was thoroughly informed about the study's objectives and procedures and provided written consent to participate. The consent form for this single participant has been submitted to the Editor during the paper submission process. We hope that this information provides a satisfactory answer to your questions regarding the ethical aspects.

Q: The methods section does not explain the evaluation measures, nor their psychometric characteristics in stroke.

A: Thank you for observing the omission of psychometric characteristics for the evaluation measures in our “Experimental design” section. We want to inform you that we have now completed this section to include a thorough discussion of the psychometric characteristics of the evaluation measures used in our study. This addition will enhance the rigor and transparency of our research, contributing significantly to the overall quality of our manuscript, see lines 320-384.

Minor issues:

Q: I recommend reorganizing the introduction to make it easier to follow the information presented. For example, the first paragraph about COVID seems out of place to me. I would first introduce the information on "Post-stroke rehabilitation strategies" and then specify the three included in the combined therapy of the case presented.Furthermore, it would indicate whether there is evidence of the use of Myofascial release (MFR) therapy in people with stroke or in neurological patients.

A: Thank you for your valuable feedback. We have reorganized the introduction as per your suggestion, starting with an introduction to "Post-stroke rehabilitation strategies" and specifying the three included in the combined therapy of the case presented. Additionally, we have supplemented the content with relevant references providing evidence of Myofascial release (MFR) therapy in people with stroke and neurological patients, see lines 187-202.  

Q: I recommend improving Table 1 to make it more visual. Similarly, Table 2 should also be reviewed.

A: Thank you for your suggestion. We have enhanced the visual presentation of Table 1 and reviewed Table 2 accordingly. 

Q: The first 5 paragraphs of the results section practically repeat the intervention without providing results.

A: Thank you for your valuable feedback. We have completed these paragraphs integrating in the text our relevant results, see lines 419-442, and lines 448-467.

Q: Regarding the conclusions, I recommend rewriting them since, except for the last paragraph, the rest repeats information or is a poorly founded conclusion as it is based on a single clinical case. In the last paragraph they talk about "community reintegration" when this result has not been studied.

A: Thank you for your observation. We have taken your recommendations into account, and we have revised the conclusions accordingly. To address your concern about community reintegration, we have now included the 7th point of the CARE guideline, which emphasizes the importance of conveying the patient's perspective and experiences in clinical case reports, see lines 578-601.

Comments on the Quality of English Language

Q: I recommend reviewing the English and simplifying the sentences to facilitate understanding. For instance, see lines 87-89 and 89-91.

A: Thank you for pointing out these complex sentences, the paragraph was updated and structured for clarity. 

Reviewer 2 Report

Comments for Submission "A Combined Rehabilitation Approach for Improving Mobility and Cardiovascular Fitness in Chronic Post-Stroke Patients: A Single-Subject Case Report"

The study provides a concise summary of the study investigating the effects of a combined rehabilitation approach on mobility and cardiovascular fitness in chronic post-stroke patients. The study outlines the primary motor recovery goals and the three rehabilitation techniques employed in the study: myofascial release therapy, ballistic strength training, and interval aerobic training. The results indicate positive outcomes in terms of improved gait speed, gait quality, and cardiovascular fitness. Overall, the report effectively conveys the main objectives, methods, and critical results of the study.

However, some minor revisions are necessary to enhance clarity and precision.

First, it would be beneficial to include specific details regarding the patient's clinical profiles, such as brain imaging scans at admission, NIHSS levels, and upper extremity functions. Any change in terms of the lesion column, or functional recovery amount between the admission and chronic stage (when patients received the PT intervention) needs to be added. This information helps readers know if the patient had reached the recovery plateau before receiving the treatment.

Besides, it would be valuable to mention any limitations or potential implications of the findings to give readers a comprehensive understanding of the study's scope. Due to the single-case design, the carry-over effect of different treatment protocols can be observed. 

Lastly, the rationale for selecting the three protocols for this patient should be better presented. 

NA

Author Response

Dear Reviewer,

We extend our gratitude for your meticulous review of our manuscript and for providing us with invaluable insights and recommendations for its enhancement. Your expertise and careful evaluation are sincerely appreciated.

We are pleased to respond to your comments and suggestions, addressing each point you raised:

Q: First, it would be beneficial to include specific details regarding the patient's clinical profiles, such as brain imaging scans at admission, NIHSS levels, and upper extremity functions. Any change in terms of the lesion column, or functional recovery amount between the admission and chronic stage (when patients received the PT intervention) needs to be added. This information helps readers know if the patient had reached the recovery plateau before receiving the treatment.

A: We appreciate your thorough review of our manuscript, specifically regarding the patient clinical profiles. We understand your request for additional details, such as brain imaging scans at admission, NIHSS levels, and upper extremity function assessments. However, the patient in our case did not receive brain imaging scans during the rehabilitation period, which limits our ability to provide this specific data. Regarding upper extremity functions, it is pertinent to note that although modest improvements were observed in left upper limb movements, particularly at the shoulder level, the patient continued to experience pronounced rigidity in the left hand. However, it is imperative to acknowledge that the primary research focus of this study centered on lower limb function and mobility, in alignment with the patient's objective of achieving medium-distance ambulation, see lines 481-504.

Q: Besides, it would be valuable to mention any limitations or potential implications of the findings to give readers a comprehensive understanding of the study's scope. Due to the single-case design, the carry-over effect of different treatment protocols can be observed. 

A: Thank you for the remark, however, based on several literature sources, stroke patients in the subacute and chronic phase of the recovery tend to experience a plateau phase. We supplemented the text of the manuscript with relevant data. Our multiple therapeutic approach was designed starting from the chronic phase of the rehabilitation, precisely because spontaneous improvement in motor function and conventional approaches tend to lead to a plateau phase in the late subacute phase, while our observed variables support the plateau phase, e.g. muscle strength in knee flexors and knee extensors, see table 4., active range of motion values for hip flexion, hip extension, knee flexion, ankle plantar flexion and dorsiflexion, all of which showed improvement in the chronic phase after our therapy. Our single subject/study offers a base for further trials with higher subject numbers, in order to statistically back-up our findings. In the late subacute phase of our study, ranging from 17 to 24 weeks, our patient reached a plateau phase: muscle spasticity was not improving – e.g. Knee flexor and Knee extensor strength, AROM measurements with goniometry, weak endurance, stagnant gait quality with proximal compensation, and reduced distal power generation. Our study clearly has limitations, however, we emphasize that our single-subject study is valuable from the perspective of evidence-based clinical application of home-based and simple intervention techniques for this type of patient. We updated the manuscript text.

Q: Lastly, the rationale for selecting the three protocols for this patient should be better presented. 

A: Thank you for your feedback. We have made the necessary additions, see lines 229-297 and 320-384.

Round 2

Reviewer 1 Report

Thank you very much for having into account my suggestions. I think that now the intervention and the case report is much more clear. There are a few minor issues that should be addressed:
- Abstract, lines 40-43, it seems that the conclusion does not coincide with the conclusion of the paper.
- Introduction: there is a lot of evidence about physical therapy and stroke, thus why it is justify to report a single case? which literature gap do you tackle? which is the purpose of the study?

Please, review english writting throughout the entire manuscript since there are some minor editing of English language required.

Author Response

Dear Reviewer,

We extend our gratitude for your meticulous review of our manuscript, both the initial review and the subsequent one, and for providing valuable insights and recommendations for its enhancement.

We are pleased to respond to your comments and suggestions, addressing each point you raised:

R1

Comments and Suggestions for Authors

Thank you very much for having into account my suggestions. I think that now the intervention and the case report is much more clear. There are a few minor issues that should be addressed:

Q: Abstract, lines 40-43, it seems that the conclusion does not coincide with the conclusion of the paper.

A: Thank you for the remark, we addressed the incoherence in our rationale and completed the manuscript for clarity in the Discussion part in lanes  574-581, and in the Conclusions part in lines 616-617 and 619-625, respectively.

Q: Introduction: there is a lot of evidence about physical therapy and stroke, thus why it is justify to report a single case? which literature gap do you tackle? which is the purpose of the study?

A: We are conscious about the standards for clinical research and the limitations of single-subject case studies, as stated in our original manuscript in lanes 508-515 in the Discussion section. However, the novel COVID-19-related stroke cases are still in need of specific, even single-case-derived information on rehabilitation approaches (see lines  60-67 in the Introduction section). To further emphasize and clarify these aspects, we supplemented the Introduction part in lines 115-117.

Comments on the Quality of English Language

Q: Please, review english writting throughout the entire manuscript since there are some minor editing of English language required.

A: Thank you for your remark. We subsequently reviewed the English language use throughout the manuscript, and typo errors and formulations were corrected, e.g. lines 284, 333, overall English formulation was updated for clarity, e.g lines 362-367, 496-498.

Reviewer 2 Report

The authors have addressed all minor points. Well Done.

Author Response

Dear Reviewer,

Thank you for your feedback and kind words. We're pleased to hear that the minor points have been successfully addressed. Your input is greatly appreciated, and we value your support in improving our work.

Sincerely,

Dr. Pál SALAMON 

Round 3

Reviewer 1 Report

After reading the last version of your manuscritp, I think now it is much clearer. Thank you for your time and effort.

English language has been improved.

Author Response

Dear Reviewer,

Thank you for your review and kind words. We're pleased to hear that the manuscript's clarity has improved and that the quality of the English language is now better. Your feedback was valuable in enhancing the manuscript, and we appreciate your time and effort in providing comments.

Sincerely,

Dr. Pal SALAMON